# A Joint Method for Wave and Wind Field Parameter Inversion Combining SAR with Wave Spectrometer Data

**Yong Wan** [1,*] , **Xiaona Zhang** [1], **Chenqing Fan** [2], **Ruozhao Qu** [3] and **Ennan Ma** [3]

[1] College of Oceanography and Space Informatics, China University of Petroleum, Qingdao 266580, China;
s21160003@s.upc.edu.cn
[2] Remote Sensing Office of the First Institute of Oceanography, Ministry of Natural Resources,
Qingdao 266061, China; fanchenqing@fio.org.cn
[3] College of Control Science and Engineering, China University of Petroleum, Qingdao 266580, China;
s19050018@s.upc.edu.cn (R.Q.); s20050037@s.upc.edu.cn (E.M.)
* Correspondence: wanyong@upc.edu.cn; Tel.: +86-150-5325-1676

**Abstract:** Synthetic aperture radar (SAR) and wave spectrometer are common methods for observing the ocean wind field and waves on the sea surface, but both have limitations. Due to the influence of velocity bunching modulation, SAR wave observation is limited by the azimuth cut-off phenomenon. Meanwhile, SAR relies on an external wind direction source, so it is difficult for SAR to observe wind fields independently. There is no azimuthal cut-off phenomenon when the spectrometer observes the sea surface, but its azimuth resolution is much lower than SAR. Combining the above characteristics, the joint inversion of SAR and wave spectrometer for sea-state parameters becomes possible. In this paper, a joint method for wind field and wave parameter inversion combining SAR with spectrometer data is proposed. In this method for wave parameter inversion, the wave spectrum of the spectrometer was used as a first-guess spectrum of SAR wave spectrum inversion, the fit wave spectrum obtained by joint inversion and the modified wave spectrum of the spectrometer were fused to form a new spectrum, and the wave parameters were calculated. For wind field parameter inversion, wind direction was obtained using a wave spectrometer, and was used as the input of the SAR wind field inversion. Wind speed was obtained using the CMOD5.N method. Collocated data from Sentinel-1 SAR, the wave spectrometer, the National Data Buoy Center (NDBC) buoy, and the European Centre for Medium-Range Weather Forecasting (ECMWF) are used to verify the proposed method. Sea parameters retrieved from the spectrometer and SAR are compared to the buoy and ECMWF data. The results show that the root mean square errors of significant wave height, mean wave period, wind direction and wind speed are 0.37 m, 1.02 s, 22.7° and 1.06 m/s with ECMWF data, and 0.35 m, 0.78 s, 18.22° and 0.92 m/s with buoy data, respectively. By comparing the inversion results with the L2 products of SAR and SWIM, it can be concluded that the inversion accuracy of the joint method is higher in the middle and low sea conditions. Therefore, the joint inversion method for wind field and wave parameters proposed in this paper has good results, which verifies the accuracy of the joint inversion method.

**Keywords:** SAR; wave spectrometer; joint inversion; wave and wind field parameters

## 1. Introduction

The ocean contains a variety of important ocean dynamic processes, such as wind field and waves on the sea surface. The wind field is the main power source of relative motion of the sea surface, and waves are a kind of small-scale wind-generated gravity wave caused by the wind field of sea surface. The measurement of the wave and wind field is important for the security of nations with coastal borders and the development of various marine resources [1,2].

At present, the main means of monitoring ocean elements are buoy observation, wave numerical model prediction and remote sensing observation [3]. Although buoy

observation has high accuracy, it cannot be observed in a wide range [4]. Wave numerical model prediction is the result of numerical calculation, not the result of field detection, and the accuracy of prediction is affected by a variety of factors. Consequently, neither buoy observation nor wave numerical model prediction is an ideal means of wave information acquisition. Sensors mainly used for remote sensing observation include SAR and wave spectrometers, etc. As an active microwave imaging sensor, SAR has its own radiation source, does not rely on external natural light and the revisit period is short. SAR is a major means of detecting the wind field and waves in a wide range with high resolution [5]. However, SAR wave observation is limited by velocity bunching modulation, resulting in a phenomenon of azimuth cut-off [6]. This phenomenon causes SAR to lose some azimuth information. Meanwhile, SAR wind field observation cannot be carried out independently, and inversion results depend on the external wind direction source. The wave spectrometer is a new type of active microwave remote sensing radar, which can detect the backscattering coefficient of the sea surface by using the small incident angle beam, then extracting the wave spectrum, wavelength, wave direction and significant wave height, and it is not limited by the phenomenon of azimuth cut-off. In addition, the wave spectrometer can obtain the wind direction by inverting the relationship between the azimuth of radar and backscattering coefficient [7], but the azimuth resolution of the wave spectrometer is much lower than SAR.

Combining the above characteristics of SAR and the wave spectrometer, it can be seen that the joint inversion of SAR and the wave spectrometer is expected to solve the problems existing in the single inversion, and may provide a new idea for ocean observation in future.

After years of development, a large number of scientists engaged in SAR wave inversion. Hasselmann et al. studied the nonlinear transformation relationship between a SAR image spectrum and a two-dimensional wave spectrum, and developed the Max Planck Institute (MPI) algorithm for wave spectrum inversion based on the SAR image spectrum [8,9]. Mastenbroek et al. and He Yijun, respectively, proposed a semi-parametric inversion method (SPRA) [10] and a parametric inversion method [11] based on the nonlinear transformation theory of Hasselmann. Engen et al. proposed a wave inversion scheme based on the cross-spectrum method, which significantly improved the accuracy of wave inversion and effectively solved the 180° ambiguity of wave propagation [12]. Schulz-Stellenfleth et al. combined the MPI algorithm and cross-spectrum algorithm to obtain a new inversion algorithm, the Partition Rescaling and Shift Algorithm (PARSA), which can eliminate the 180° ambiguity in the wave propagation direction. At the same time, this algorithm can also obtain higher inversion accuracy than the cross-spectrum algorithm [13]. However, the above methods do not solve the problem of azimuth cut-off in SAR wave detection. Secondly, some scholars have made important contributions to the study of wave spectrum detection using the wave spectrometer. Jackson et al. demonstrated the feasibility of using airborne and space-borne real aperture radar (RAR) to measure the wave spectrum, and discussed the linear relationship between the signal modulation spectrum and the slope spectrum [14]. Jackson, Hauser et al. designed and developed the radar wave spectrometers ROWS and RESSAC, which were renamed STORM subsequently [15,16]. Hauser et al. used the space-borne radar SWIMSAT to research and monitor surface waves, and concluded that SWIMSAT could measure wave spectral information with wavelengths greater than 70 m [17]. Tison et al. studied the ability of SWIM wave spectrum inversion by means of simulation, and established the inversion simulation platform of the SWIM wave direction spectrum [18]. Based on the merit and demerit of SAR and the wave spectrometer, Ren Lin et al. used SAR and airborne Real Aperture Radar (RAR) to conduct a simulation study on detecting the direction spectrum with a low incidence angle, and the results show that RAR could detect waves that can be measured by SAR [19]. Ren Lin et al. also proposed a method for retrieving the wave spectrum from SAR and wave spectrometer data [20].

Moreover, many scholars also engaged in wind field inversion. At a medium incidence angle, Geophysical Model Functions (GMF) are the most frequently used by researchers

when SAR was used to retrieve wind field parameters. This method is an empirical model based on statistical data of SAR and wind speed. At present, GMF applicable to C-band mainly includes CMOD4, CMOD-IFR2, CMOD5 and CMOD5.N. Stoffelen et al. tested and verified the ability of the CMOD4 function to fit conic surfaces [21]. Quilfen et al. processed the backscattering coefficient data through the CMOD-IFR2 algorithm and obtained the wind vector [22]. Hersbach et al. derived a CMOD5 model that could be applied in high wind speeds based on CMOD4 [23], and they also used CMOD5.N to correct the errors in CMOD5 [24]. However, the above methods cannot avoid the problem that SAR cannot detect the wind field independently. At a small incidence angle, Chu Xiaoqing adopted a large number of Precipitation Radar (PR) data to establish an empirical relationship between the backscattering coefficient, incidence angle and wind speed [25]. Wang Xiaochen used ECMWF data to discuss the obvious anisotropy of the backscattering coefficient at the sea surface when the incident angle of the wave spectrometer was 4–18° [26]. Li Peng concluded that SWIM is suitable for wind speed and wind direction inversion [27].

According to the above analysis of SAR and the wave spectrometer, the key problems in the SAR and wave spectrometer inversion are as follows: SAR is limited by azimuth cut-off. [6] and cannot observe the wind field independently. The wave spectrometer has low resolution in azimuth.

In view of the above key problems, the sea-state parameters are retrieved by SAR and the wave spectrometer in this paper. This method not only makes up for the shortcomings of SAR with azimuth cut-off and dependence on wind direction data, but also solves the low resolution problem of the wave spectrometer. Finally, the significant wave height, mean wave period, wind direction and wind speed obtained by the joint inversion are compared with NDBC and ERA-5 data, respectively, so that the accuracy of the joint method for sea-state parameter inversion combining SAR with wave spectrometer data can be verified.

The remainder of this article is organized as follows. The datasets and methods used in this study are described in Section 2. The feasible explanation of our findings is discussed in Section 3, followed by the final remarks in Section 4.

## 2. Data Sources and Methods

This chapter mainly introduces the data sources and methods used in the research.

### 2.1. Data Sources

#### 2.1.1. Sentinel-1 SAR Data

Sentinel-1 consists of two satellites. On 3 April 2014, ESA launched Sentinel-1A, which updates its data every 12 days. After the launch of Sentinel-1B on 26 April 2016, the two satellites can cooperate with the earth observation, and the revisit period of the satellite can be shortened to 6 days. Sentinel-1 SAR has four imaging modes: Stripmap (SM), Interferometric Wide Swath (IW), Extra Wide Swath (EW) and Wave Mode (WV). Sentinel-1 data used in this paper are C-band VV-polarized WV mode data. WV mode is the default imaging mode used by SAR satellites for sea surface observation, and SAR images can be obtained alternately at incident angles of 23° and 36° along the orbit.

#### 2.1.2. CFOSAT SWIM Data

The China France Ocean Satellite (CFOSAT) is jointly developed by China and France. China is responsible for developing the radar scatterometer, while France is responsible for developing the wave spectrometer. Launched in Jiuquan on 29 October 2018, the wave spectrometer carried by Surface Waves Investigation and Monitoring (SWIM) is the world's first satellite-borne wave spectrometer. SWIM is a radar for detecting wave information combined with altimeter and small incident angle multi-beam real-aperture imaging radar, and it can provide all-day and near-all-weather wave measurements around the world. The wave spectrometer is equipped with multiple beams of 0°, 2°, 4°, 6°, 8° and 10°. The significant wave height is retrieved from wave spectrometer data using the narrow nadir beam (0°), as for altimeters. Other beams are used to measure the direction spectrum

information of waves, and finally the accuracy of significant wave height measurements is better than 0.5 m. The wavelengths that can be detected are 70–500 m [28].

Among them, the time and space matching interval of collocated SAR and SWIM data are 1.5 h and 50 km, respectively.

### 2.1.3. ECMWF ERA5 Reanalysis Data

The ERA-5 reanalysis data are a dataset that combines model data with observations from around the world. Since 1979, it has provided researchers with many oceans' atmospheric data, including significant wave height, mean wave period, wind speed and direction at 10 m above the sea surface, and so on. The maximum spatial resolution can reach $0.125° \times 0.125°$ [29]. In this paper, the time and space matching interval of collocated SAR, SWIM and ECMWF data are 1 h and 12.5 km, respectively.

### 2.1.4. Buoy Data

Ocean buoys are the main means of monitoring and forecasting marine disasters around the world. At present, the development of ocean buoy technology in many countries has been perfected, and buoy data is recognized as the most accurate. Therefore, the significant wave height, mean wave period, wind speed and wind direction provided by the NDBC are used as real data in this paper. Since the NDBC buoy provides wind speed data of 5 m above sea level [30], the wind speed obtained by the COMD5.N function in this study is 10 m above sea level. Hence, the wind speed provided by the NDBC buoy needs to be converted. The conversion formula is as follows [31]:

$$U_z = U_{z_m} \frac{\ln(z/z_0)}{\ln(z_m/z_0)} \tag{1}$$

where $U_z$ represents the wind speed at the height of $z$ meters above the sea surface; $U_{z_m}$ represents the wind speed at the height of $z_m$ meters above the sea surface; $z_0$ represents the length of the roughness.

The time and space matching interval of collocated SAR, SWIM and NDBC data are 1 h and 50 km, respectively. The accuracy of sea-state parameter inversion can be verified by using the matched buoy data. The above SAR and SWIM data are the most basic data sources for the study, and ECMWF ERA-5 and NDBC data are used as the verification data sources.

Figure 1a shows the distribution of the matched data (S-1 SAR WV and CFOSAT SWIM) during a revisit (212 sets), where each blue block represents the location of a set of matched data. Figure 1b shows the positions of the SAR image and buoy, and is a glimpse of the WV mode SAR image of Sentinel-1B. The SAR image was acquired at 14:16 on 22 December 2020. The white circle and red line in the figure represent the position of the buoy and SWIM (15:25, 22 December 2020) data in the area covered by the SAR image.

### 2.2. Joint Inversion Method for Wave Parameters

In this paper, the inversion of wave parameters is carried out through the method of a combined wave spectrum; the specific steps are as follows: Firstly, the wave spectrum of the spectrometer is input into the MPI method as a first-guess spectrum; the fit wave spectrum and the fit SAR spectrum are output. Secondly, the significant wave height calculated by the fit wave spectrum and the narrow nadir beam, respectively, are used to modify the wave spectrum of the spectrometer, and the updated wave spectrum is input into the MPI method again as the first-guess spectrum for iteration. Finally, the fit wave spectrum of SAR is spliced with the modified wave spectrum of the spectrometer. According to the spliced wave spectrum, significant wave height and mean wave period can be obtained. The inversion process is shown in Figure 2.

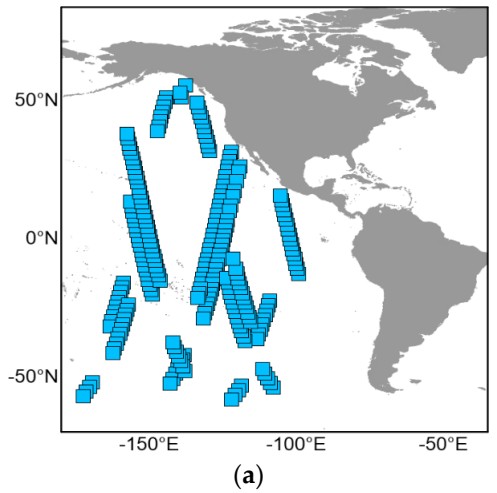

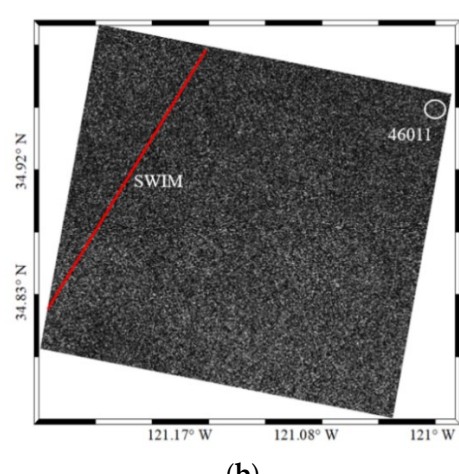

(**a**)                                                 (**b**)

**Figure 1.** Data matching: (**a**) Distribution of matched data (S-1 SAR WV and CFOSAT SWIM) over a revisit period. (**b**) Position of Buoy 46011 and SWIM in SAR image.

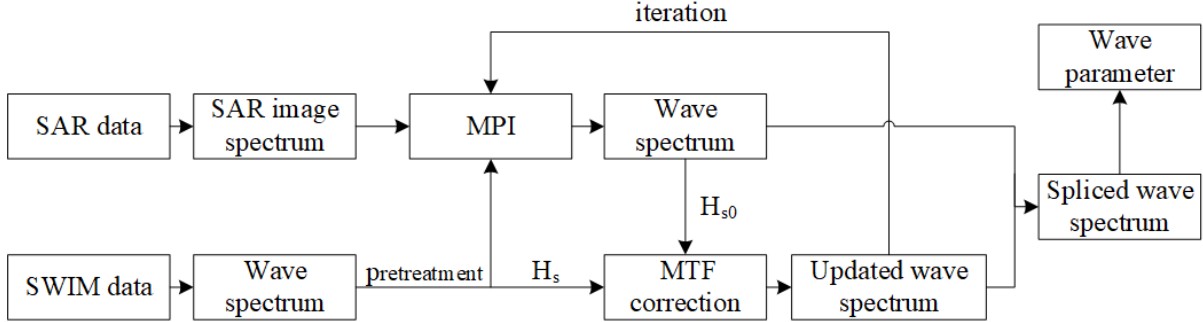

**Figure 2.** Flowchart of wave joint inversion method for SAR and wave spectrometer.

2.2.1. Pretreatment of the SWIM Wave Spectrum

The wave spectrum, as shown in Figure 3 can be provided in SWIM data, which is taken as a first-guess input in the MPI method. Obviously, we know that the first-guess spectrum required for SAR wave inversion is a two-dimensional wavenumber spectrum, but the wave spectrum of SWIM is a two-dimensional wavenumber direction spectrum, so it is necessary to convert the wave spectrum of SWIM. The conversion relationship between the two is shown in Equation (2) [32].

$$F(kx, ky) = F(k, \varphi)\frac{1}{k} \tag{2}$$

where $F(kx, ky)$ is the wavenumber spectrum, $F(k, \varphi)$ is the wavenumber direction spectrum, $k$ is the wavenumber, and $kx = k \sin \varphi$, $ky = k \cos \varphi$.

The wavenumber spectrum obtained by the above method has a horizontal direction of due east and a vertical direction of due north. However, the horizontal direction of the SAR wave spectrum is the range direction, and the vertical direction is the azimuth direction. Therefore, in this paper, the SWIM wave spectrum is inserted into the SAR wave spectrum; the first-guess spectrum that can constitute SAR wave inversion is obtained at the end.

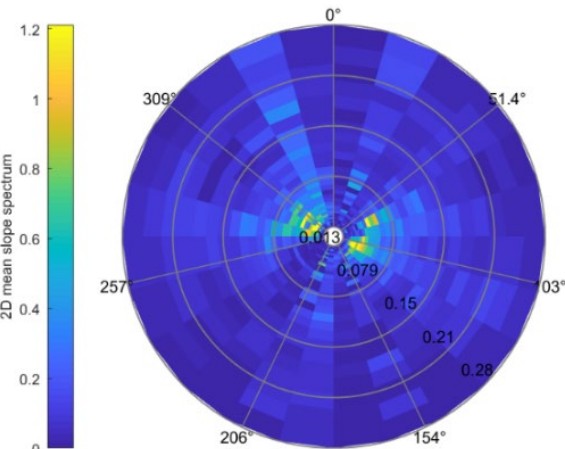

**Figure 3.** Wave spectrum of wave spectrometer.

### 2.2.2. MPI Method

It is difficult to invert the wave direction spectrum from the observed SAR image directly because the transformation relationship between the wave direction spectrum and the SAR image spectrum is nonlinear. Hasselmann et al. established the nonlinear integral transformation relation between the SAR image spectrum and the two-dimensional wave spectrum, and then obtained the MPI method that required the input of the first-guess spectrum [8,9]. Although this method requires a first-guess spectrum, its inversion information is relatively complete and accurate under the same conditions. Therefore, this paper uses the MPI method to invert the wave spectrum.

When inverting the wave spectrum, firstly, we input the SAR image spectrum and the first-guess spectrum provided by the wave spectrometer. Then, the forward mapping relationship is used to obtain the simulated SAR image spectrum. Next, we substitute the simulated SAR image spectrum and the SAR image spectrum into the value function, and determine whether the iterative process converges [33].

The value function J is defined as follows [9]:

$$
J = \int [P(k) - \hat{P}(k)]^2 \hat{P}(k)dk + \mu \int \left\{ \frac{[F(k) - \hat{F}(k)]}{[B + \hat{F}(k)]} \right\}^2 dk \tag{3}
$$

where $\hat{F}(k)$ is the first-guess spectrum and $F(k)$ is the fit wave spectrum when J takes the minimum value. $\hat{P}(k)$ and $P(k)$ are the SAR image spectrum and the fit SAR spectrum, respectively. $\mu$ is the weight coefficient, and the small positive number B avoids the value function from being infinite when $\hat{F}(k) = 0$. The wave spectrum results are shown in Figure 4.

In this paper, a SAR image of 256 × 256 pixels at the intersection of collocated SAR and wave spectrometer data is selected to generate the observed SAR spectrum by a two-dimensional Fourier transform. The observed SAR spectrum and the first-guess spectrum are input into MPI method to output the fit wave spectrum and the fit SAR spectrum. The inversion results of wave spectrum are shown in Figure 5.

Wave parameters can be calculated from the first-order moment, second-order moment, and higher-order moments of the wave frequency spectrum $S(\omega)$ of the fit wave spectrum. The calculation equation for inverted significant wave height is as follows [33]:

$$
H_{s0} = 4\sqrt{m_0} = 4\sqrt{\int S(\omega)d\omega} \tag{4}
$$

where $m_0$ is the first-order moment.

The calculation equation for the inverted mean wave period is as follows [33]:

$$T_m = 2\pi \sqrt{\frac{m_0}{m_2}} = 2\pi \sqrt{\frac{\int S(\omega)d\omega}{\int \omega^2 S(\omega)d\omega}} \tag{5}$$

where $m_0$ and $m_2$ are the first-order moment and the second-order moment, respectively.

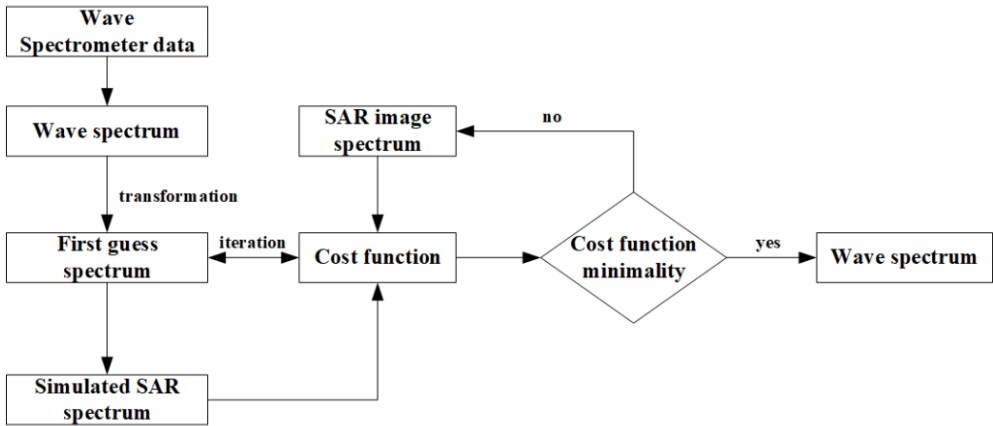

**Figure 4.** Flowchart of MPI method of different first-guess spectra.

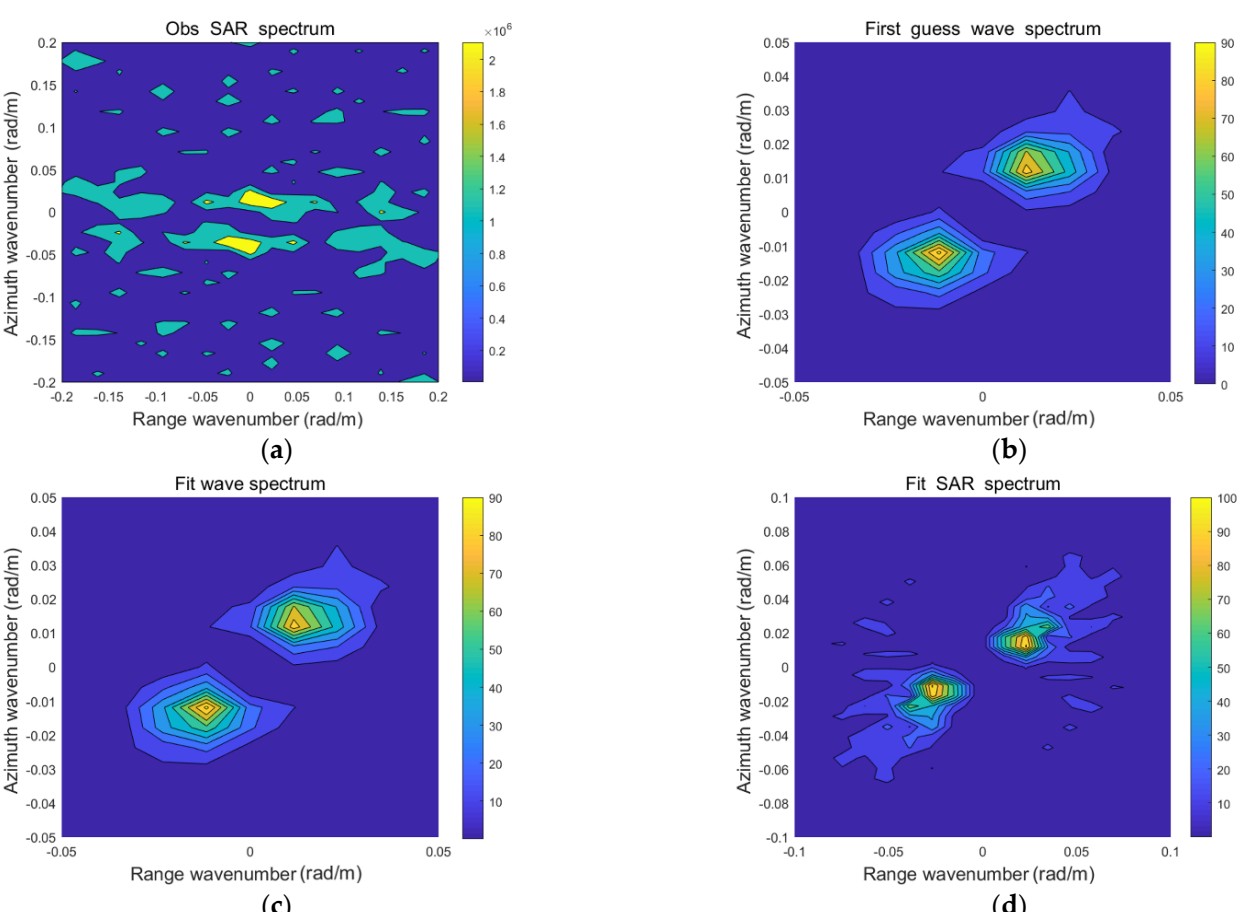

**Figure 5.** Wave spectrum inversion results in 256 × 256 pixels: (**a**) Observation of SAR spectrum; (**b**) First-guess spectrum; (**c**) Fit wave spectrum; (**d**) Fit SAR spectrum.

Finally, as shown in Figure 5a,d, the shortwave component of the fit SAR spectrum after inversion is supplemented, so it can be seen that the joint method for wave and wind

field parameter inversion combining SAR with wave spectrometer data proposed in this paper can be used as a method to supplement the SAR azimuth cut-off.

### 2.2.3. MTF Correction and Wave Spectrum Splicing

In this paper, 150–500 m is assumed as the wavelength range of joint inversion for wave parameters, due to cut-off wavelength mainly occurring in this range [9]. However, when the wavelength is around 500 m, there may be a pseudo-peak in the wave spectrum of the wave spectrometer, which has a great influence on the calculation of significant wave height. Therefore, this paper does not consider the wave spectrum data of the spectrometer whose wavelength is around 500 m, and selects the intersection of 150–400 m as the wavelength range of the joint inversion for wave parameters. The inversion results of SAR are considered more accurate in this range. Then, a new Modulation Transfer Function (MTF0) can be obtained. After MTF0 is applied to the whole wave number range (70–400 m), a new wave spectrum of the wave spectrometer can be generated.

The transfer function between the modulation spectrum $P_m(k, \varphi)$ and the slope spectrum $k^2 F(k, \varphi)$ is given by this Equation (2) [19]:

$$P_m(k, \varphi) = \frac{\sqrt{2\pi}}{L_y} \alpha^2(\theta) k^2 F(k, \varphi) \tag{6}$$

where $\theta$ is the beam incidence angle, $\varphi$ is the observation direction, $\alpha$ is the sensitivity coefficient, $L_y$ is the width of the azimuth for the radar footprint and $F(k, \varphi)$ is the wave height spectrum.

Unlike SAR, Equation (6) shows that the transfer function between the modulation spectrum $P_m$ and the wave slope spectrum $k^2 F(k, \varphi)$ is linear, so the wave spectrometer is not limited by azimuth cut-off [6].

The width of the azimuth $L_y$ for the radar footprint in Equation (6) is defined as follows [19]:

$$L_y = r \frac{B\varphi}{2\sqrt{2 \ln 2}} \tag{7}$$

where r is the radial distance of the beam center and 3 dB beam width (unidirectional) in the azimuth direction of the antenna.

According to Equation (6), the modulation transfer function MTF1 of the wave spectrometer is:

$$\mathrm{MTF1}(\theta, \varphi) = \frac{\sqrt{2\pi}}{L_y} \alpha^2(\theta) \tag{8}$$

According to the SWIM product's user guide, the significant wave height $H_s$ of the wave spectrometer can be extracted from the beam at the bottom ($0°$ beam):

$$H_s = 4\sqrt{\iint k F(k, \varphi) dk d\varphi} \tag{9}$$

Therefore, according to Equations (6), (8) and (9), the relationship between MTF1 and the significant wave height $H_s$ of the wave spectrometer can be seen:

$$\mathrm{MTF1} = \frac{16\sqrt{\iint \frac{P_m(k,\varphi)}{k} dk d\varphi}}{H_s{}^2} \tag{10}$$

Due to the higher azimuth resolution of SAR, the measurement results are more accurate and precise. Therefore, in this section, the significant wave height $H_{s0}$ calculated by the fit wave spectrum and the bottom $H_s$ are used to correct the wave spectrum of the wave spectrometer, and we make the significant wave height $H_{s1} = H_{s0}$ that was calculated by the updated wave spectrum. In other words, the updated modulation function MTF0

can be obtained. As shown in Equation (11), the relationship between MTF0, $H_{s1}$ and $H_{s0}$ can be known.

$$\text{MTF0} = \frac{16\sqrt{\iint \frac{P_m(k,\varphi)}{k} dk d\varphi}}{H_{s0}{}^2} = \frac{H_s{}^2}{H_{s0}{}^2}\text{MTF1} \tag{11}$$

Applying MTF0 to the whole wave number range, the updated wave spectrum can be obtained from Equation (12).

$$S(k,\varphi) = \frac{H_s{}^2}{H_{s0}{}^2} S_{meas}(k,\varphi) \tag{12}$$

where $S_{meas}(k,\varphi)$ is the initial wave spectrum of the wave spectrometer.

The modified wave spectrum is input into the MPI method as the first guess to obtain the fit wave spectrum again. The updated wave spectrum of the wave spectrometer within the wavelength range of 70–150 m is intercepted, the intercepted wave spectrum is spliced with the fit wave spectrum (wavelength above 150 m), and finally the fused wave spectrum is formed. The wave spectrum after splicing is shown in Figure 6. Finally, the significant wave height and mean wave period are obtained.

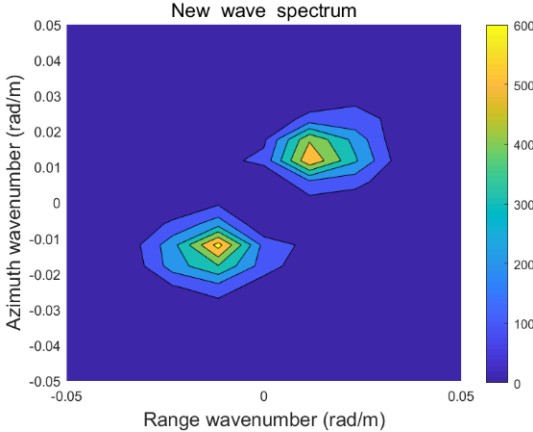

**Figure 6.** The wave spectrum after splicing.

### 2.3. Joint Inversion Method for Wind Field Parameters

In this paper, the wind direction of the wave spectrometer is retrieved first, and then the wind direction is input into the SAR wind speed inversion system. Finally, the wind speed is retrieved by the CMOD5.N method. This method does not need to carry out wind direction inversion alone, and does not rely on third-party wind direction data, except SAR and the wave spectrometer, so it can carry out a large range of high-resolution detection, and the inversion of wind speed has a higher accuracy.

#### 2.3.1. Wind Direction Inversion by Wave Spectrometer

In this section, the change of the backscattering coefficient with the radar azimuth in the range of 0–360° can be obtained through wave spectrometer data. According to the research of Wang Xiaochen [26], when the incident angle of the airborne wave spectrometer is 4–18°, the azimuth distribution of the backscattering coefficient shows an obvious double peak trend. It is determined that the azimuth direction of the wave peak is downwind and upwind, and the anisotropy and asymmetry of the backscattering coefficient increase with the increase of the incident angle. Therefore, according to the above conclusions, the relative wind direction angle is defined as $\varphi_{\text{wind}}$, where $\varphi_{\text{wind}}^*$ represents the absolute wind direction, that is, the direction from which the wind blows, and $\varphi_{\text{radar}}$ represents the radar azimuth. Thus, the scatter diagram of the backscattering coefficient of the wave spectrometer at different incident angles varies with the relative wind direction, as shown in Figure 7.

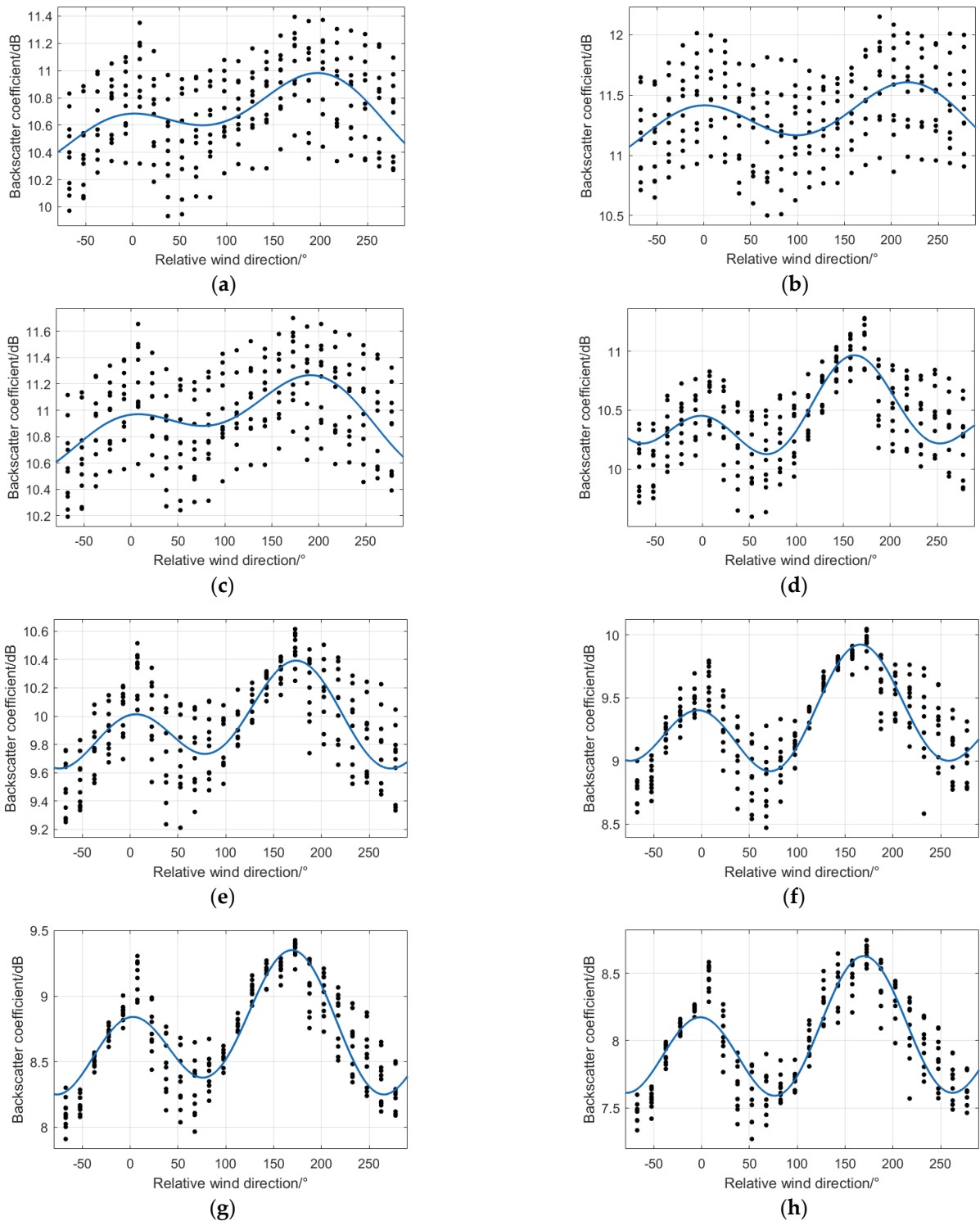

**Figure 7.** Azimuthal distribution at different incident angles: (**a**) 4°; (**b**) 5°; (**c**) 6°; (**d**) 7°, (**e**) 8°; (**f**) 9°; (**g**) 10°; (**h**) 11°.

When the SWIM incident angle is greater than 4°, the azimuthal distribution of the backscattering coefficient shows a bimodal trend, and with the increase in the incident angle, the bimodal trend is more obvious, representing stronger wind direction information. Therefore, this paper uses the incident angle of SWIM at 11° to look for the corresponding radar azimuth $\varphi_{radar}$ when the backscattering coefficient reaches its maximum value. According to the literature [26,27], when the backscattering coefficient of SWIM is the

maximum, the relative azimuth $\varphi_{\text{wind}}^* = 0°$ (upwind direction) and $\varphi_{\text{wind}}^* = 180°$ (downwind direction). Finally, the value of the wind direction parameter can be obtained from $\varphi_{\text{wind}} = \varphi_{\text{radar}} + \varphi_{\text{wind}}^*$.

### 2.3.2. Wind Speed Inversion

This section adopts the CMOD5.N method to retrieve the wind speed based on the geophysical model function, which can obtain wind speed information according to the relationship between wind direction, incident angle and backscattering coefficient. Relative wind direction is the angle between the wind direction and the SAR viewing direction, where wind direction is obtained from the inversion of the wave spectrometer, and the SAR viewing direction can be calculated by the longitude and latitude information of the SAR image vertex. The function equation of CMOD5.N is as follows [24]:

$$\sigma_0 = B_0(1 + B_1\cos\varphi + B_2\cos2\varphi)^{1.6} \tag{13}$$

where $\sigma_0$ is the backscattering coefficient, $\varphi$ is the relative wind direction, $B_0$, $B_1$ and $B_2$ are the functions of wind speed and incident angle.

In CMOD5.N, wind speed, relative wind direction and incident angle are independent variables, and the backscattering coefficient is a dependent variable, so wind speed cannot be obtained directly. In this section, wind speed inversion is realized by traversing 0–30 m/s wind speed. Finally, wind speed values of SAR and the wave spectrometer can be obtained through joint inversion [8].

## 3. Results and Discussion

### 3.1. Comparison of Inversion Results with ERA5 Data

In this section, the 212 sets of data of the Sentinel-1 SAR WV mode and space-borne wave spectrometer SWIM are used to compare the inversion results with the sea parameters of the ERA5 data; the Root Mean Square Error (RMSE) and Bias is used to evaluate the precision about the joint method for sea-state parameter inversion.

Wind direction is retrieved by the wave spectrometer; wind speed, significant wave height and mean wave period obtained by joint inversion of SAR and the wave spectrometer are compared with the ERA-5 data, and the comparison results are shown in Figure 8.

It can be seen from the above figure that the RMSE of significant wave height, mean wave period, wind direction, wind speed and ERA-5 data obtained by the joint inversion of SAR and the wave spectrometer are 0.37 m, 1.02 s, 22.77° and 1.06 m/s, respectively.

### 3.2. Comparison of Joint Inversion Results with Buoy Data

This section uses 10 sets of the collocated Sentinel-1 SAR WV mode and SWIM data, and the 10 scenes data were matched with buoy data, respectively, to verify the accuracy of the joint inversion results. The RMSE and Bias were also used to evaluate the accuracy of the joint method for sea-state parameter inversion.

The significant wave height, mean wave period, wind direction and wind speed obtained by joint inversion of SAR and the spectrometer are compared with the sea state parameters provided by buoy data, and the comparison results are shown in Figure 9.

It can be seen from the above figure that the RMSE of significant wave height, mean wave period, wind direction, wind speed and buoy data obtained by the joint inversion of SAR and the wave spectrometer are 0.35 m, 0.78 s, 18.22° and 0.92 m/s, respectively.

### 3.3. Comparison of Joint Inversion Results with L2 Level Product

The data are divided from one revisited period into three groups based on wind speed range. Ten groups of data were randomly selected from three groups of data in different wind speed ranges, respectively. The comparison results of the proposed joint inversion method with L2 level products of SAR and SWIM are shown in Table 1.

**Table 1.** The RMSE between the parameters of the three methods and ERA5 were obtained under different wind speed ranges.

| The Range of Wind Speed (m/s) | SAR | SWIM | SAR and SWIM |
|---|---|---|---|
| Ws < 6 | Hs: 0.64 m<br>Ws: 1.46 m/s | 0.38 m<br>1.60 m/s | 0.46 m<br>1.18 m/s |
| 6 < Ws < 10 | Hs: 0.30 m<br>Ws: 1.10 m/s | 0.22 m<br>1.23 m/s | 0.37 m<br>1.20 m/s |
| Ws > 10 | Hs: 1.27 m<br>Ws: 1.77 m/s | 0.28 m<br>0.66 m/s | 0.70 m<br>1.53 m/s |

As can be seen in Table 1: (1) when wind speed is less than 6 m/s, although the inversion accuracy of the joint method is close to that of the SAR product, the inversion accuracy of the wind speed is the highest, so the inversion effect of the joint method is the best within this range. (2) when the wind speed is between 6 m/s and 10 m/s, the parameter inversion accuracy of the three methods is very close. (3) when the wind speed is greater than 10 m/s, the parameter inversion accuracy of the SWIM product is the highest, followed by the joint method. It can be seen from the above that the combined method has high accuracy in the inversion of middle and low sea conditions, so it has high research value.

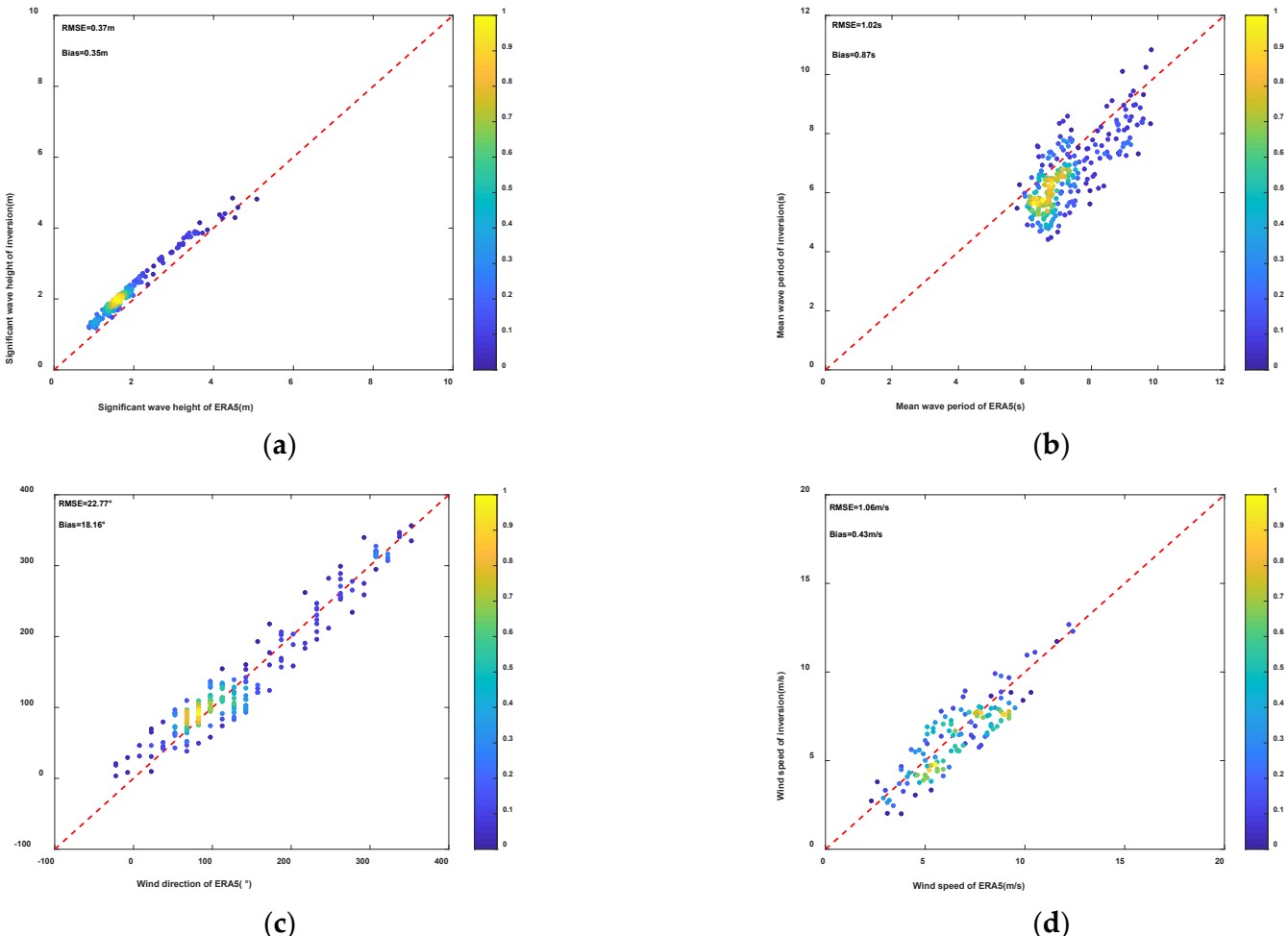

**Figure 8.** Comparison of the inversion results with the ECMWF ERA5 data: (**a**) Significant wave height; (**b**) Mean wave period (**c**) Wind direction; (**d**) Wind speed.

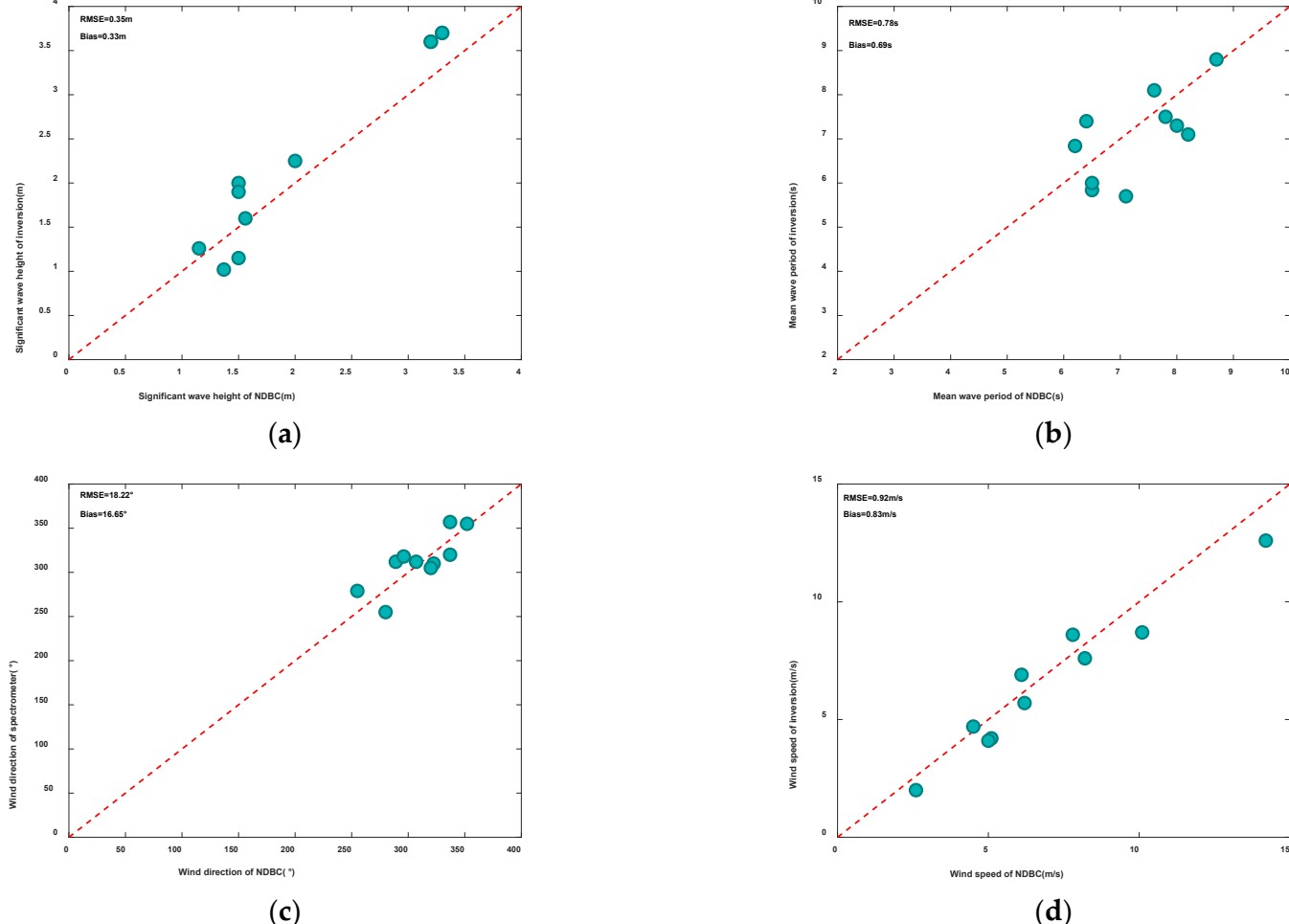

**Figure 9.** Comparison of the inversion results with buoy data: (**a**) Significant wave height; (**b**) Mean wave period; (**c**) Wind direction; (**d**) Wind speed.

## 4. Conclusions

This paper proposes a joint method for wave and wind field parameter inversion combining SAR with wave spectrometer data. In this method for wave parameter inversion, firstly, the wave spectrum of the wave spectrometer was used as a first-guess spectrum. Then, the first-guess spectrum and the observed SAR spectrum were input into the MPI method to obtain the fit wave spectrum. Next, MTF was used to modify the wave spectrum of the wave spectrometer, and the modified wave spectrum was re-input into the MPI method as the first-guess spectrum to obtain the updated fit wave spectrum. Finally, the spliced wave spectrum was obtained by combining the fit wave spectrum with the modified wave spectrum of the wave spectrometer. For wind field parameter inversion, the wind direction of the wave spectrometer was used as the input of the CMOD5.N function, and wind speed was output. The sea state parameters by inversion were compared with ERA5 and buoy data to evaluate the precision of the joint inversion method. Meanwhile, the parameters of the joint method were compared with L2 level products of SAR and SWIM.

The conclusions are as follows:

(1) For wave parameter inversion, this paper proposes a new method for wave parameter inversion combining SAR with wave spectrometer data. This method makes up for the defect of azimuth cut-off in SAR. The RMSE of the wave parameters and ERA-5 and buoy data show that this joint method for wave parameter inversion is feasible.

(2) For wind field parameter inversion, this paper uses the wind direction of the wave spectrometer as the input of the CMOD5.N function, which can be independent of

the external data, except for SAR and the wave spectrometer. The RMSE of wind field parameters and ERA-5 and buoy data show that this joint method for wind field parameter inversion is feasible.

(3) Compared with L2 level SAR and SWIM product parameters, the joint method has better applicability in the middle and low sea conditions, so this method has a high research value.

Finally, the joint inversion of sea state parameters does not rely on third-party data information and can be used for high-resolution detection in a wide range. The results show that the inversion results have good accuracy when the wind speed is less than 15 m/s. In future work, we will continue to improve the method to increase the inversion accuracy in high sea states. In addition, the method described in this paper can provide reference for future research into the joint inversion method of SAR and the wave spectrometer.

**Author Contributions:** Data curation, Y.W. and X.Z.; investigation, C.F.; resources, C.F., R.Q. and E.M.; writing—original draft preparation, X.Z.; writing—review and editing, Y.W. All authors have read and agreed to the published version of the manuscript.

**Funding:** This research was funded by the National Natural Science Foundation of China, grant number No. 61931025 and the National Key R&D Program of China, grant number No. 2017YFC1405600.

**Acknowledgments:** Thanks to the National Data Buoy Center for buoy data, the European Space Agency for SENTINEL-1 SAR data, the China–France Ocean satellite CFOSAT data satellite service system for providing SWIM data and the European Center for Medium and Long-term Weather Forecasts for the ECMWF ERA5 data.

**Conflicts of Interest:** The authors declare no conflict of interest.

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
