# Peer review of "A Joint Method for Wave and Wind Field Parameter Inversion Combining SAR with Wave Spectrometer Data"

_remotesensing, doi:10.3390/rs14153601_

Round 1
Reviewer 1 Report
At present, SAR and SWIM onboard CFOSAT are the only two kinds of spaceborne radars detecting directional ocean waves at a global scale, and both of them are capable of observing ocean winds. This paper presents a wind/wave retrieving method for jointly using these two spaceborne radars. The topic is new and interesting.
My remarks are:
1. First of all, the authors should present a map of Sentinel-1 and CFOSAT/SWIM collocations during a revisit period. This is of importance because the authors have to tell the readers synchronous or quasi-synchronous observations from these two radars indeed exist, how much and where they are. The enough spatiotemporal distribution number of the collocation is the prerequisite of the proposed joint inversion methodology.
For this point, I suggest the authors refer to the first paper regarding the Sentinel-1A/B Wave Mode and CFOSAT/SWIM collocations. (Wang et al., 2022)
2. The feature of SAR and SWIM for waves is the directional spectral (Jiang et al., 2022 and doi: 10.1109/tgrs.2021.3110952.). So only consider the integral spectral parameters (SWH for instance ) is not enough, you should validate the 2D spectra or partitioned SWH of your results.
3. section "2.2.1. Pretreatment of the SWIM wave spectrum"
The SWIM products (as shown in Fig. 3) are wave slope spectra, which are needed to converted to the energy spectra before using equation (1)
4. Fig 10(d). When comparing satellite derived wind speed (U10) and buoy data, one should convert the buoy wind speeds to the equivalent neutral winds at 10 m using the method like COARE algorithm (Fairall et al., 2003).
5. The ref cited in the manuscript "Wang, X.C. Study on wave spectrum inversion method of airborne spectrometer. MA Thesis, China University of Petroleum (East China), Qingdao, China, 2016." is not publicly available. Since this is quite important in section 2.3.1., I suggest the authors uploaded this thesis to somewhere (like dropbox or onedrive) for readers.
6.Both Sentinel-1A/B Wave Mode and CFOSAT/SWIM have official products (see Johnsen et al., 2020 ; Hauser et al., 2021, respectively). Thus, it is interesting to see whether the proposed jointly inversion has better performance than the official SAR and SWIM L2 products.
References:
(1) Wang, H., Mouche, A., Husson, R., Chapron, B., Yang, J., Liu, J., Ren, L., 2022. Quantifying Uncertainties in the Partitioned Swell Heights Observed From CFOSAT SWIM and Sentinel-1 SAR via Triple Collocation. IEEE Transactions on Geoscience and Remote Sensing, 60, 1–16. doi: 10.1109/tgrs.2022.3179511.
(2) H. Jiang, A. Mironov, L. Ren, A. V. Babanin, J. Wang, and L. Mu, “Validation of Wave Spectral Partitions From SWIM Instrument On-Board CFOSAT Against In Situ Data,” IEEE Trans. Geosci. Remote Sens., pp. 1–13, 2022, doi: 10.1109/tgrs.2021.3110952.
(3)H. Wang, A. Mouche, R. Husson, and B. Chapron, “Indian Ocean Crossing Swells: New Insights from "fireworks” Perspective Using Envisat Advanced Synthetic Aperture Radar,” Remote Sens., vol. 13, no. 4, p. 670, Feb. 2021.
(4) C. W. Fairall, E. F. Bradley, J. E. Hare, A. A. Grachev, and J. B. Edson,
“Bulk parameterization of air–sea fluxes: Updates and verification for
the COARE algorithm,” J. Climate, vol. 16, no. 4, pp. 571–591,
Feb. 2003.
(5) Harald Johnsen, Romain Husson, 2020. Sentinel-1 Ocean Swell Wave Spectral Algorithm Definition. ESA
(6) Hauser, D., et al., 2021. New Observations From the SWIM Radar On-Board CFOSAT: Instrument Validation and Ocean Wave Measurement Assessment. IEEE Transactions on Geoscience and Remote Sensing, 1–22.
Author Response
Dear Editor and Reviewers,
Thank you and reviewer for giving constructive comments for our manuscript. Our manuscript entitled ‘‘A joint method for wave and wind field parameter inversion combined SAR with wave spectrometer data’’ has been revised in response to the comments. Thank you for the opportunity to resubmit this revision manuscript for possible publication in the journal.
We have given serious consideration to Editor’s and Reviewer’s comments and suggestions and have revised the manuscript carefully. In order to make your review easy, we have reworked the content and format of the manuscript and have marked all modified sections by using red text in the file “Revised Manuscript with changes Marked”. Manuscript revisions are addressed in detail below.
Reviewer:
General comment: At present, SAR and SWIM onboard CFOSAT are the only two kinds of spaceborne radars detecting directional ocean waves at a global scale, and both of them are capable of observing ocean winds. This paper presents a wind/wave retrieving method for jointly using these two spaceborne radars. The topic is new and interesting.
Revision detail:
Thank you very much for your valuable suggestions. We really appreciate your efforts in reviewing our manuscript. We have revised the manuscript according to your comments. The point-by-point responses are detailed below.
Specific Comments:
- First of all, the authors should present a map of Sentinel-1 and CFOSAT/SWIM collocations during a revisit period. This is of importance because the authors have to tell the readers synchronous or quasi-synchronous observations from these two radars indeed exist, how much and where they are. The enough spatiotemporal distribution number of the collocation is the prerequisite of the proposed joint inversion methodology.
Revision detail:
Thank you for your professional comment. As there are many NDBC buoy data in the Pacific coast, the study area is the Pacific Ocean in this manuscript. On this basis, we add the distribution map of matching data (SAR and wave spectrometer) in one revisit period of S1-A and S1-B as shown in Figure 1a, each blue block represents a set of matched data and it contains 212 sets of matching data.
- The feature of SAR and SWIM for waves is the directional spectral (Jiang et al., 2022 and doi: 10.1109/tgrs.2021.3110952.). So only consider the integral spectral parameters (SWH for instance ) is not enough, you should validate the 2D spectra or partitioned SWH of your results.
Revision detail:
Thank you for your professional comment. Since the main content of this manuscript is the inversion of wave and wind field parameters, we did not involve the study of two-dimensional spectrum and partition SWH, but I think the idea is very interesting. Please allow us to continue to promote this part of work in the follow-up work.
- section "2.2.1. Pretreatment of the SWIM wave spectrum"
The SWIM products (as shown in Fig. 3) are wave slope spectra, which are needed to converted to the energy spectra before using equation (1).
Revision detail:
Thank you very much for your careful review. After receiving your comments, we carefully checked and studied the content in this section. After consulting the data, the two-dimensional power spectrum, the wavenumber direction spectrum and wavenumber spectrum formula transformation relationship between 1 and 2. And the slope spectrum obtained from spectrometer data is the two-dimensional wavenumber direction spectrum, we want to finally get to wave spectrum is two-dimensional wavenumber spectrum, so according to the discussion above we believe that the slope spectrum don't need to be converted to energy spectrum. We are very sorry that we do not have a thorough understanding of this problem and know that we are not good enough. Therefore, we sincerely hope that you can spend your precious time to criticize and correct. Thank you again for your professional advice [1].
Note: Formulas 1 and 2 are in the uploaded PDF file
[1] Yijun He, Zhongfeng Qiu, Biao Zhang. Ocean Wave Observation Technology [M]. Science Press, 2015.
- Fig 10(d). When comparing satellite derived wind speed (U10) and buoy data, one should convert the buoy wind speeds to the equivalent neutral winds at 10 m using the method like COARE algorithm (Fairall et al., 2003).
Revision detail:
Thank you for your professional question. Since the NDBC buoy provides wind speed data of 5 meters above sea level, the wind speed obtained by COMD5.N function in this study is 10 meters above sea level. Therefore, we convert the wind speed provided by the NDBC buoy according to equation (1).
- The ref cited in the manuscript "Wang, X.C. Study on wave spectrum inversion method of airborne spectrometer. MA Thesis, China University of Petroleum (East China), Qingdao, China, 2016." is not publicly available. Since this is quite important in section 2.3.1., I suggest the authors uploaded this thesis to somewhere (like dropbox or onedrive) for readers.
Revision detail:
Thank you for your careful review. Users can search this thesis on CNKI platform, but we are sorry that we may not be able to upload it to the platform (like dropbox or onedrive) due to possible copyright problems. If you would like to view this manuscript, please allow us to send it to the journal editor or to you.
- Both Sentinel-1A/B Wave Mode and CFOSAT/SWIM have official products (see Johnsen et al., 2020; Hauser et al., 2021, respectively). Thus, it is interesting to see whether the proposed jointly inversion has better performance than the official SAR and SWIM L2 products.
Revision detail:
Thank you for your professional suggestion. I think the questions you raised are very useful, which can make my research more meaningful. Therefore, I divided the data from one revisited period into three groups based on wind speed range. Ten groups of data were randomly selected from the three groups, respectively. Here we evaluate the inversion accuracy of the three methods with the root mean square error of significant wave height and wind speed and ERA5 data.
As can be seen from Table 1, (1) when wind speed is less than 6m/s, although the inversion accuracy of the combined method is close to that of the SAR product, the inversion accuracy of the wind speed is the highest, so the inversion effect of the combined method is the best within this range. (2) When the wind speed is between 6m/s and 10m/s, the parameter inversion accuracy of the three methods is very close. (3) When the wind speed is greater than 10m/s, the parameters inversion accuracy of SWIM product is the highest, followed by the combined method. It can be seen from the above that the combined method has high accuracy in the inversion of middle and low sea conditions, so it has high research value.
We would like to express special thanks to the editor and reviewer for their constructive comments on this manuscript again. Please tell us some shortcoming again about this manuscript when reviewing. We hope this manuscript will be accepted by Remote Sensing.
Thanks again, we look forward to your reply.
Best Regards.
Sincerely,
Dr. Yong Wan

Reviewer 2 Report
This manuscript introduces a joint method for wave and wind field parameter inversion combined SAR with wave spectrometer data. It is an interesting work and has practical reference value for future joint observation of SAR and wave spectrometer satellites application. I suggest this manuscript can be accepted after minor revision.
Detailed comments:
(1) "Letter" is suitable for a short article, so I suggest using "Paper" in this article.
(2) As shown in Equation 11, "Pm" subscript is too large, please modify it.
(3) As shown in Figure 5, the cut-off wavelength of SAR has been compensated. I am not sure how the cut-off phenomenon is compensated, please add a description of this problem.
(4) Page 10, I do not know the relationship between the downwind direction, upwind direction of the spectrometer and the inversion wind direction, please mark downwind direction and upwind direction in Figure 7 or explain their relationship clearly in words.
(5) As shown in Figure 7, how is wind direction obtained from wave spectrometer data? Please add a description of the inversion mechanism about wind direction.
(6) When some abbreviations are first mentioned in the manuscript, please give a specific description. For example: NDBC; ECMWF, etc.
(7) As shown in Figure 8-9, it is too thin to use only the root mean square error to evaluate the inversion accuracy. I suggest adding other parameters to evaluate the accuracy, such as bias, etc.
(8) There are 30 groups of SAR and wave spectrometer data matching with ERA5 data in this paper, and I suggest to increase this data appropriately.
Author Response
Dear Editor and Reviewers,
Thank you and reviewer for giving constructive comments for our manuscript. Our manuscript entitled ‘‘A joint method for wave and wind field parameter inversion combined SAR with wave spectrometer data’’ has been revised in response to the comments. Thank you for the opportunity to resubmit this revision manuscript for possible publication in the journal.
We have given serious consideration to Editor’s and Reviewer’s comments and suggestions and have revised the manuscript carefully. In order to make your review easy, we have reworked the content and format of the manuscript and have marked all modified sections by using red text in the file “Revised Manuscript with changes Marked”. Manuscript revisions are addressed in detail below.
Reviewer:
General comment: This manuscript introduces a joint method for wave and wind field parameter inversion combined SAR with wave spectrometer data. It is an interesting work and has practical reference value for future joint observation of SAR and wave spectrometer satellites application. I suggest this manuscript can be accepted after minor revision.
Revision detail:
Thank you very much for your professional suggestions and approbation of our manuscript. We have tried our best to modify and improve our manuscript according to your comment, the point-by-point responses are shown in our revised manuscript.
Specific Comments:
- "Letter" is suitable for a short article, so I suggest using "Paper" in this article.
Revision detail:
Thank you for your careful review. We have rechecked the paper and adjusted these the phrase and sentence with ‘Letter’.
- As shown in Equation 11, "Pm" subscript is too large, please modify it.
Revision detail:
Sorry for our carelessness. Thank you very much for your careful review. We have rechecked the manuscript and corrected the Equation 11.
- As shown in Figure 5, the cut-off wavelength of SAR has been compensated. I am not sure how the cut-off phenomenon is compensated, please add a description of this problem.
Revision detail:
Thank you very much for your careful review. Unlike SAR, Equation 2 shows that the transfer function between the modulation spectrum and the wave slope spectrum is linear, so the wave spectrometer does not have the azimuth cut-off phenomenon. Because we use the wave spectrum provided by wave spectrometer as the first guess spectrum of MPI method, the cut-off wavelength in the fit SAR spectrum is compensated in the end.
- Page 10, I do not know the relationship between the downwind direction, upwind direction of the spectrometer and the inversion wind direction, please mark downwind direction and upwind direction in Figure 7 or explain their relationship clearly in words.
Revision detail:
Thank you for your professional comment. According to literature [25, 26], when the backscattering coefficient of SWIM is the maximum, the relative azimuth φ*wind=180° (upwind direction) and φ*wind=0°(downwind direction). We have explained the mechanism of the inversion mechanism about wind direction in detail and added it after Figure 7.
- As shown in Figure 7, how is wind direction obtained from wave spectrometer data? Please add a description of the inversion mechanism about wind direction.
Revision detail:
Thank you for your extremely useful comment. According to literature [25, 26], when the backscattering coefficient of SWIM is the maximum, the relative azimuth φ*wind=180° (upwind direction), φ*wind=0°(downwind direction) and the azimuth of the radar φradar. can be obtained from the spectrometer data. Finally, the value of wind direction parameter can be obtained from φwind=φ*wind +φradar. We have explained the relationship downwind direction, upwind direction and wind direction in detail and added it after Figure 7.
- When some abbreviations are first mentioned in the manuscript, please give a specific description. For example: NDBC; ECMWF, etc.
Revision detail:
Thank you for your careful review. All the abbreviations have been described when some abbreviations are first mentioned in our revised manuscript.
- As shown in Figure 8-9, it is too thin to use only the root mean square error to evaluate the inversion accuracy. I suggest adding other parameters to evaluate the accuracy, such as bias, etc.
Revision detail:
Thank you for your professional suggestion. In Figure 7 and Figure 8, we evaluate the inversion accuracy of parameters by using both bias and root mean square error.
- There are 30 groups of SAR and wave spectrometer data matching with ERA5 data in this paper, and I suggest to increase this data appropriately.
Revision detail:
Thank you for your professional comment. The enough spatiotemporal distribution number of the collocation is the prerequisite of the proposed joint inversion methodology. Therefore, we evaluate the applicability of the joint approach using 212 sets of matching data from a review period in this manuscript.
We would like to express special thanks to the editor and reviewer for their constructive comments on this manuscript again. Please tell us some shortcoming again about this manuscript when reviewing. We hope this manuscript will be accepted by Remote Sensing.
Thanks again, we look forward to your reply.
Best Regards.
Sincerely,
Dr. Yong Wan

Reviewer 3 Report
Comments:
· Page 1, line 13: it is not clear what do you mean by azimuth cut-off phenomenon and by velocity bunching modulation. It is highly suggested to introduce in an analytical way the problem that the paper wants to solve.
· Page 1, line 16: “although wave spectrometer doesn’t exist azimuth cut-off”. This is not correct English. As a general comment I think that it is ok if the paper is not written is perfect English (I’m also non-native English speaker as it is clear by my poor English). In this paper, however, I think that the language barrier is preventing the reader to properly understand the paper. I highly suggest doing a professional review of the English of the entire paper.
· Page 1, line 20-23: the word “wave” is repeated 9 times in 3 lines. I suggest to use a synonym or somehow simplify the sentence.
· Page 2, line 52: “all day long” is not scientific. Revisit frequency or repetition interval should be used.
· Page 2, line 53: again, it is not clear what is azimuth cut-off. References to previous paper are also suggested.
· Page 2, line 67: “domestic and foreign” depends on your country. You should write for all the scientists, not only for the Chinese community. A more proper sentence could be “After years of development, a large number of scientists engaged in SAR wave inversion”.
· Page 3, line 110-113: they key problems that you are mentioning are not clear from the references. “As can be seen from the above inversion methods…”. I personally can’t see the problem from the cited methods.
· Page 3, line 121: please add the sections index like “In section 2 we discuss…”
· Page 4: the Section 2.2 in generally confusing. Block diagrams are not clear, they waste a lot of space in the page. My suggestion is to write a good block diagram and then explain each block of the diagram with a section. Here 2.2 seems a partial explanation of the whole procedure, then there are more details in 2.2.1,2.2.2, etc. is it correct?
· Page 5, line 192-193: the word “wave” is used again 7 times in 2 lines of text.
· Page 6, equation 3: why the first term of the equation is different from the second one? Why there is no denominator in the first?
· General comment: all the equations are poorly written: integrals should be bigger. The MDPI template is very good (LaTex even better) with equations.
· General comments: plot are also generally poor. For example, in Figure 5 a much better visualization can be done by interpolating the spectra (simple 2D FFT with an higher number of frequency points). This can eventually also improve the results.
· Equations in general seems to come from nowhere. Are there other works that use these formulae? If yes, please cite them. If no, please explain better the formulae.
Author Response
Dear Editor and Reviewers,
Thank you and reviewer for giving constructive comments for our manuscript. Our manuscript entitled ‘‘A joint method for wave and wind field parameter inversion combined SAR with wave spectrometer data’’ has been revised in response to the comments. Thank you for the opportunity to resubmit this revision manuscript for possible publication in the journal.
We have given serious consideration to Editor’s and Reviewer’s comments and suggestions and have revised the manuscript carefully. In order to make your review easy, we have reworked the content and format of the manuscript and have marked all modified sections by using red text in the file “Revised Manuscript with changes Marked”. Manuscript revisions are addressed in detail below.
Reviewer:
General comment: all the equations are poorly written: integrals should be bigger. The MDPI template is very good (LaTex even better) with equations.
Revision detail:
Sorry for our carelessness. Thank you for your careful review. We have checked and modified all the equations in the manuscript.
General comments: plot are also generally poor. For example, in Figure 5 a much better visualization can be done by interpolating the spectra (simple 2D FFT with an higher number of frequency points). This can eventually also improve the results.
Revision detail:
Thank you for your professional suggestion. We have modified some of the plots in the manuscript. Because figure 5 is the four process diagrams generated in MPI method, it is mainly to make the SAR azimuth cut-off wavelength be compensated by comparing Figure 5a and Figure 5d in the manuscript. Although we did not use interpolation spectroscopy, we replaced the SAR image of 128*128 pixels with that of 256*256 pixels based on your suggestion, which can make our wave-number interval smaller and the inversion result more accurate. This also allows for better visualization.
General comments: Equations in general seems to come from nowhere. Are there other works that use these formulae? If yes, please cite them. If no, please explain better the formulae.
Revision detail:
Thank you for your professional suggestion. We have marked the source of the formulas in this manuscript. Formula 9 is the calculation formula of wave height of sky bottom point of spectrometer, which is from SWIM products users guide.
Specific Comments:
- Page 1, line 13: it is not clear what do you mean by azimuth cut-off phenomenon and by velocity bunching modulation. It is highly suggested to introduce in an analytical way the problem that the manuscript wants to solve.
Revision detail:
Thank you for your professional suggestion. Due to the nonlinear effect of velocity bunching modulation, there is azimuth cut-off phenomenon in SAR wave observation, which is the inherent defect of SAR wave observation. As a result, SAR cannot observe the complete wave information, and the accuracy of wave observation is limited, which greatly restricts the ability of SAR wave observation and the realization of commercial application. As the abstract should not be too many words, we have introduced it in the introduction with references [6, 9].
- Page 1, line 16: “although wave spectrometer doesn’t exist azimuth cut-off”. This is not correct English. As a general comment I think that it is ok if the paper is not written is perfect English (I’m also non-native English speaker as it is clear by my poor English). In this paper, however, I think that the language barrier is preventing the reader to properly understand the paper. I highly suggest doing a professional review of the English of the entire paper.
Revision detail:
Sorry for our carelessness. Thank you for your professional comments. We have carefully checked the English of the manuscript and modified it. Thank you again for your advice.
- Page 1, line 20-23: the word “wave” is repeated 9 times in 3 lines. I suggest to use a synonym or somehow simplify the sentence.
Revision detail:
Thank you very much for your careful review. Due to some of them are specialized words, and each “wave” stands for a different meaning, “wave parameters” represent effective wave height, average wave period and other parameters, “ wave spectrometer” represents a type of radar, and “wave spectrum” is mainly used to calculate wave parameters. Therefore, we are sorry that we cannot change all of them, but we have checked it carefully and modified it as much as possible. Thank you again for your advice.
- Page 2, line 52: “all day long” is not scientific. Revisit frequency or repetition interval should be used.
Revision detail:
Thank you for your extremely useful comment. We have reviewed the manuscript and adjusted the unscientific sentences and vocabulary.
- Page 2, line 53: again, it is not clear what is azimuth cut-off. References to previous paper are also suggested.
Revision detail:
Thank you for your professional suggestion. We have introduced it in the introduction with references. We hope it will give you a better understanding of the azimuth cut-off phenomenon of SAR.
- Page 2, line 67: “domestic and foreign” depends on your country. You should write for all the scientists, not only for the Chinese community. A more proper sentence could be “After years of development, a large number of scientists engaged in SAR wave inversion”.
Revision detail:
Thank you for your very careful review. We have corrected the grammatical mistakes in this manuscript as you suggested.
- Page 3, line 110-113: they key problems that you are mentioning are not clear from the references. “As can be seen from the above inversion methods…”. I personally can’t see the problem from the cited methods.
Revision detail:
Thank you for your professional suggestion. The azimuth cut-off problem of SAR and the low azimuth resolution of spectrometer are inherent defects, and the inversion method mentioned in this paper cannot avoid the above problems. This is also a key issue that we want to address. Therefore, based on the existing inversion methods of SAR and spectrometer, we establish two radar joint inversion methods for wave and wind field parameters. We have introduced the problems with SAR and wave spectrometer before introducing the key issues, hoping to give you a better understanding of the key issues addressed in this article. Thank you again for your thoughtful advice.
- Page 3, line 121: please add the sections index like “In section 2 we discuss…”.
Revision detail:
Thank you very much for your helpful comment. We have added content according to your suggestion in our revised manuscript. We hope that this content will help you and readers understand the chapter distribution of this paper more quickly.
- Page 4: the Section 2.2 in generally confusing. Block diagrams are not clear, they waste a lot of space in the page. My suggestion is to write a good block diagram and then explain each block of the diagram with a section. Here 2.2 seems a partial explanation of the whole procedure, then there are more details in 2.2.1, 2.2.2, etc. is it correct?
Revision detail:
We are very sorry that we did not notice this problem. Thank you for your careful suggestion. Section 2.2.1 mainly introduces the pretreatment of SWIM wave spectrum. The processed wave spectrum can be used as an input of MPI method. Section 2.2.2 mainly introduces the MPI method that the wave spectrum of the spectrometer as first guess spectrum. We have modified it in the manuscript and we believe that it can make you better understand the method proposed in this manuscript.
- Page 5, line 192-193: the word “wave” is used again 7 times in 2 lines of text.
Revision detail:
Thank you very much for your careful review. Due to some of them are specialized words, and each “wave” stands for a different meaning, “two dimensional wavenumber spectrum” and “wavenumber direction spectrum” represent the horizontal and vertical form of the SWIM wave spectrum, “ wave spectrometer” represents a type of radar, and “wave spectrum” is mainly used to calculate wave parameters. Therefore, we are sorry that we cannot change all of them, but we have checked it carefully and modified it as much as possible. Thank you again for your advice.
- Page 6, equation 3: why the first term of the equation is different from the second one? Why there is no denominator in the first?
Revision detail:
Thank you very much for your professional suggestion. This formula is the cost function established by Hasselmann, K and Hasselmann, S according to the characteristics of SAR in sea surface detection [9]. Due to the 180° ambiguity of SAR images and the information loss caused by azimuth cut-off, the only normal inversion form of forward mapping relationship does not exist. The standard way to deal with this kind of uncertain inversion problem is to introduce the regular term to use the extra information in the first guess spectrum F (k). The two terms of the formula have different information to obtain, so the two terms are different.
We would like to express special thanks to the editor and reviewer for their constructive comments on this manuscript again. Please tell us some shortcoming again about this manuscript when reviewing. We hope this manuscript will be accepted by Remote Sensing.
Thanks again, we look forward to your reply.
Best Regards.
Sincerely,
Dr. Yong Wan

Round 2
Reviewer 1 Report
No further questions. Thanks for the response from the authors.
Reviewer 3 Report
I’ve no further comments